# Integrating Neuroimaging and Genetics via Contrastive Learning for Working Memory

Pranav Nadigapu Suresh
Computer Science
Georgia State University
Atlanta, USA
0000-0001-9167-3718

Behnam Kazemivash
Computer Science
Georgia State University
Atlanta, USA
bkazemivash1@gsu.edu

Dawn M Jensen
TReNDS
Georgia State University
Atlanta, USA
djensen2@gsu.edu

Vince Calhoun
TReNDS
Georgis State university
Atlanta, USA
vcalhoun@gsu.edu

Jingyu Liu
Computer Science
Georgia State University
Atlanta, USA
jliu75@gsu.edu

*Abstract—* **Understanding working memory's genetic and neural bases is crucial for advancing cognitive neuroscience and identifying biomarkers for cognitive impairments, particularly in the older population. This study integrates SNP and neuroimaging data from the UK biobank to improve the classification of high vs. low working memory capacity and reveal genetic factors associated with brain structure. 1060 SNPs belonging to Protein-Protein Interaction networks of amyloid precursor protein and Aβ of Alzheimer's disease were integrated with latent features of whole brain gray matter density, extracted by a pre-trained CNN, via supervised contrastive learning. Our model effectively extracts latent representations of both modalities through enhancing genetic-imaging relation within individuals and within working memory groups, in contrast to across individuals and groups. Features derived from contrastive learning outperformed other baseline models in terms of classification. Sparse canonical correlation analysis was applied to the latent representations and uncovered significantly related genetic variants and brain regions. Genetic components highlight SNPs in genes FYN, RPL28, MAPT, enriched in the pathways of dendrite and synapse, among others. The linked brain regions support the cerebellum and striatum's role in cognitive functions. These findings provide new insights into the genetic and neural mechanisms underlying working memory, potentially guiding future research and therapeutic strategies for cognitive impairment.**

*Keywords—Working memory, Genetic variants, Neuroimaging, Contrastive learning, Sparse Canonical Correlation Analysis*

## I. INTRODUCTION

WORKING MEMORY (WM), a fundamental cognitive function, enables individuals to hold and manipulate information over short periods, essential for reasoning, learning, and decision-making. It is influenced by both genetic and neural factors. Understanding its genetic and neural basis can help identify biomarkers for cognitive impairment in neurodegenerative diseases [1] in the old population, as working memory declines in normal aging, Mild Cognitive Impairment and Alzheimer's disease (AD) [2]. Furthermore, working memory is also one of earliest key symptoms in AD [3], making it a good phenotypical outcome to study for genetic mutations associated with AD.

Previous research has established various links between genetic variants and brain structure/function that underpin working memory performance. As reviewed in [4] one of the most studied genes is COMT (Catechol-O-Methyltransferase), which encodes an enzyme that degrades dopamine in the prefrontal cortex, a region critical for working memory [5]. Yet its genetic variants may code normal working memory variation in the population. Specifically interested in the working memory decline in the old population, we leverage the knowledge of protein-protein interaction networks in AD [6], and conduct a focused study on genetics involved in pathogenesis of AD with brain structure serving WM function. Among many genes, for instance, ADAM10 (a member of the A Disintegrin And Metalloproteinase (ADAM) family) is known to be involved in the cleavage of amyloid precursor protein, a key process of AD pathogenesis, and also helps normal synaptic functions and hippocampal neurogenesis [7].

Neuroimaging studies have identified specific brain regions whose function or structural variations are crucial for WM. Functional MRI (fMRI) studies consistently show that the prefrontal cortex, particularly the dorsolateral prefrontal cortex (DLPFC), is heavily involved in WM tasks [8]. Recent studies have highlighted the role of the cerebellum in WM, especially the left cerebellum being implicated in verbal WM tasks [9-11]. A transdiagnostic study has revealed consistent patterns of dysfunction in the prefrontal and parietal cortices, as well as cerebellum, across various psychiatric and neurological diagnoses [12]. In parallel, structural MRI (sMRI) studies echo the findings of fMRI, where DLPFC surface area independently contributes to WM performance [13], and grey matter volumes in the inferior frontal and cerebellum are associated with WM across age groups [14].

The integration of genetic and imaging data provides deeper insights into how genetic variants influence brain, thereby affecting WM. Heck et al. performed genome-wide gene set enrichment analyses in multiple data sets, young and aged, and identified the voltage-gated cation channel activity gene set was linked to WM-related tasks and parietal cortex and the cerebellum [15]. Previous works have linked a set of single nucleotide polymorphisms (SNP) to gray matter alterations in the frontal region underlying WM deficits in adults and adolescents with attention-deficit/hyperactivity disorder. The SNPs highlighted MEF2C, CADM2, and CADPS2, relevant for modulating neuronal substrates underlying high-level cognition [16].

Both traditional data fusion approaches, such as Canonical Correlation Analysis (CCA) [17, 18] and parallel

This research was funded by NIH grant RF1AG063153 to Calhoun/Liu.

independent component analyses [19], and deep learning based approaches [20] have been implemented for integrating neuroimaging data and genetics. Yet due to heterogeneous characteristics of imaging and genetics, it is a still changeling task to effectively integrate datasets. Contrastive learning techniques have recently emerged as powerful tools for multi-modal data integration. These methods learn shared representations from different data types by maximizing their agreement, making them particularly suitable for tasks involving genetic and imaging data. A recent study by Taleb et al. (2022) introduced ContIG, a self-supervised multimodal contrastive learning framework for medical imaging with genetics. ContIG effectively learns joint representations by contrasting positive pairs (genetic and imaging data from the same subject) against negative pairs (data from different subjects), and subsequently enhances the performance of prediction tasks [21].

In this study, we leverage the strength of both CCA and contrastive learning for integration, along with transfer learning Convolutional Neural Network (CNN) and MLP for latent feature extraction and build a three-stage imaging guided SNP representation model for classification of WM capacity. The contribution of this project includes: 1) a transfer learning component from brain aging to WM for neuroimaging feature extraction, 2) a multi-modal contrastive learning approach that integrates genetic and imaging data to capture their complex relationships, 3) a sCCA interpretation for the learned representations to identify significant imaging and genetic components with shared variance. The findings could provide new insights into the biological pathways for both risk genetics and brain structure involved in WM, identify potential biomarkers for cognitive decline and impairment.

## II. MATERIALS AND METHODS

In this section we first introduce our cohort and data. Then we detail the proposed novel model, followed by baseline models for comparison. Finally, we explain the post analyses for model and results interpretation.

### A. Cohort

UK Biobank [22] contains de-identified data of a million UK participants and over 40,000 participants with brain MRI and genetic data. Informed consent was obtained from all participants, and the study was approved by the North-West Multi-center Research Ethics Committee. 26,534 subjects participated in the WM assessment by maximum digits remembered correctly during a backward digit span task, ranging from 2 digits to 12 digits with distribution as shown in Fig. 1. We selected a subpopulation of 5469 participants from two groups i) participants with memory scores ranging from 2 to 5 and ii) participants with scores in the range of 9 to 12. The segregation was based on the Miller's Law of clinical psychology which states that individuals, on average, can hold about $7 \pm 2$ items at a time in their WM [23]. We adjusted based on the distribution of our dataset to $7 \pm 1$ as the average capacity. Thus, group I is termed as low WM capacity and group II is high WM capacity. Among them, 4995 had both MRI and genetic data for our analyses, with 3192 in low WM group (age: 55.95±7.58, 1317 male), and 1803 in high WM group (age: 42.35±7.08, 1034 male).

### B. MRI Preprocessing

T1-weighted MRI images collected from three centers with identical scanners were segmented into six types of tissues (gray matter, white matter, etc.) using SPM 12 with default TPM template and modulated option, and normalized into Montreal Neurological Institute space, and resliced and smoothed with a 6×6×6mm3 Gaussian kernel. Details of pipeline can be seen in [14]. Each image has a voxels matrix

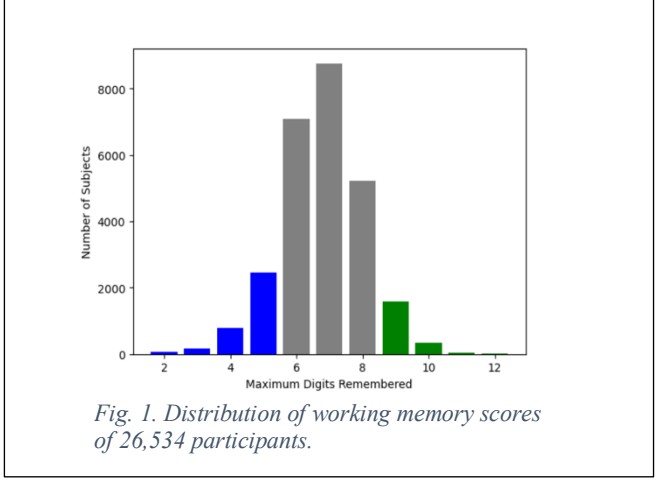

*Fig. 1. Distribution of working memory scores of 26,534 participants.*

of 128×141×128. Further quality control was conducted to retain individual images with a correlation larger than 0.9 with the averaged gray matter image.

### C. Single Nucleotide Polymorphisms (SNP) Preprocessing

The genomic data after imputation [24] were first filtered with imputation $r^2 \leq 0.3$, missingness rate $\geq 5\%$, minor allele frequency $\leq 0.01$, Hardy–Weinberg Equilibrium p value $\leq 1 \times 10\text{-}6$. Then, a subset of SNPs was selected based on expression quantitative trait loci (eQTL) for brain tissue published by PsychENCODE [24], i.e., SNPs regulating gene expression in brain. Further, to increase the likelihood of identifying the biologically relevant genes we selected SNPs that are also part of the Protein-Protein Interaction network of amyloid precursor protein and Abeta of Alzheimer's disease[25]., yielding 1060 SNPs. Each SNP was coded as 0,1, and 2 reflecting number of minor alleles.

### D. Contrastive Genetic-Neuroimaging Integration (CGNI)

We propose a three-stage imaging genetic integration framework as shown in Fig. 2. The input includes whole brain gray matter images and 1060 SNPs. The output is twofold: one is the classification of WM group, and the other is associated with latent representations of imaging and genetic data. Stage 1 is to extract imaging latent representations using transfer learning. Stage 2 is to extract genetic representations guided by the imaging representations via contrastive learning. Finally, Stage 3 combines these representations to perform WM classification. The details of each stage are provided in the following sections.

#### 1) Imaging Representation Learning Using Transfer Learning

Stage 1 shows the model to extract imaging representation. First, we pre-trained a 3D CNN model for brain age prediction task following the work of [26], using a

large sample size of 39,755. The 3D CNN architecture is composed of five convolutional blocks and each block has a 3D convolutional layer, a batch normalization layer, followed by a max pool layer, and a ReLU. The convolutional layers used 32, 64, 128, 256, 256 channels, respectively, with stride of one and no padding was used. The output images from the

neurons, respectively. The output layer is of size 100, denoted as $z_g^i = e_g(x_g^i) \in \mathbb{R}^{100}$.

For every $i^{th}$ pair of imaging representation and genetic data, $(h_v^i, x_g^i)$, $(z_v^i, z_g^i)$ are representations from the encoders $(e_v, e_g)$, in a batch of **b** samples. We constrain our batch

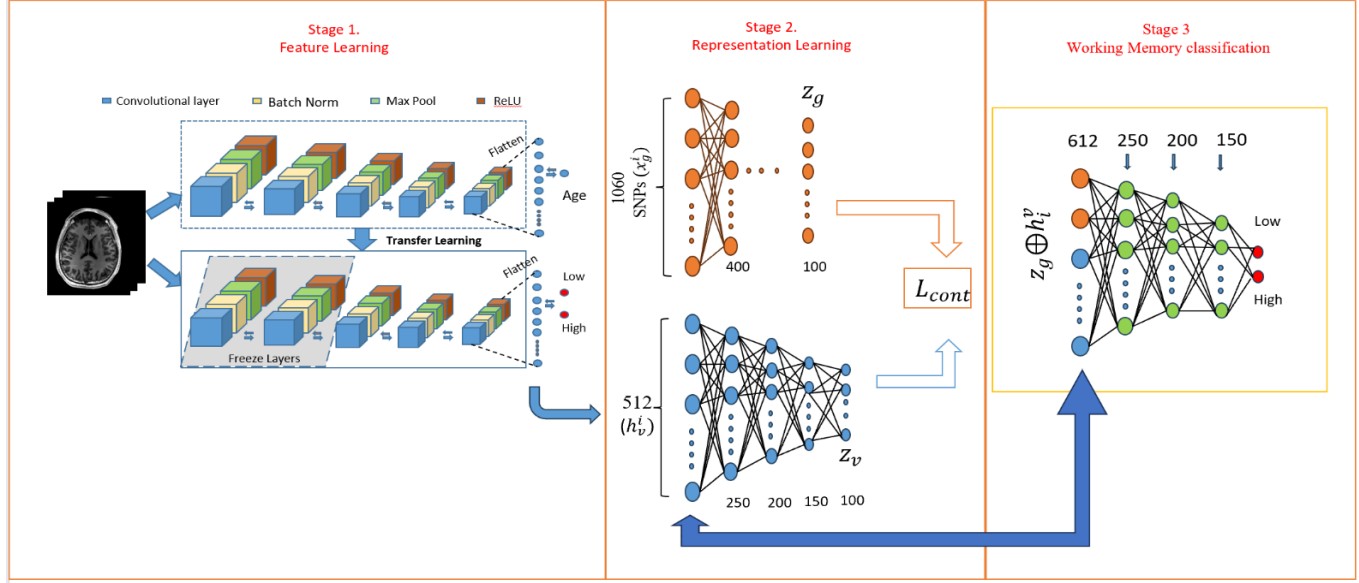

Fig. 2. Schematic illustration for the steps of our proposed method. In Stage 1, we perform feature learning of the images using Convolutional Neural Networks. In stage 2, we learn representations of the genetic modality guided by the imaging features via contrastive learning. In stage 3, we finally combine the representations from both the modalities and perform classification of working

final convolutional block are flattened and passed to the fully connected layer which takes 1024 inputs and outputs the predicted brain age. The fully connected layer has a dropout rate of 0.2.

Using transfer learning combined with heuristic self-transfer-training [27] method, the best trained brain age model was used to initialize the classification model for the prediction of the WM. We denote the flattened outputs of the final convolutional block of the 3D CNN model as $h_v^i = f_v(x_v^i)$, where $x_v^i$ is the imaging modality of the participant i $\in \{1, \dots ,4995\}$.

*2) Imaging Guided SNP Representation Learning via Contrastive Learning*

Each individual sample, $X^i$, where i $\in \{1, \dots ,4995\}$, has an imaging modality, $x_v^i$, a genetic modality, $x_g^i$, and a label $y^i \in \{low = 0, high = 1\}$. We train our contrastive model in batches of size b, where b >1. Our model, as shown in stage 2 of Fig. 2, is comprised of two encoders, one for each modality.

The encoders transform each modality into respective representations using the contrastive loss functions described in the following section. For imaging encoder, denoted as $e_v$, we use the output from the 3D CNN of the baseline model (see section D.i), $h_v^i$, as the input. The imaging encoder input layer takes 512 inputs and has three hidden layers. The size of each hidden layer is 250, 200, and 150 neurons, respectively. The output layer is of size 100, denoted as $z_v^i = e_v(h_v^i) \in \mathbb{R}^{100}$. Similarly, the genetic encoder, denoted as $e_g$, is a fully-connected neural network with 1060 SNPs as input and has five hidden layers of size 400, 600, 400, 300, and 150

selection such that $|\{y^i = 0\}| > 0$, and $|\{y^i = 1\}| > 0$, to ensure at least one sample from each label. We then define our loss function using following loss terms where $\tau$ is a temperature parameter that scales the embeddings to control the range of the dot product, and $P(i) \equiv \{p \in \{1 \dots \boldsymbol{b}\} \setminus i : y_p = y_i\}$

In (1)—imaging-to-genetics, for every imaging modality $(h_v^i)$ in a batch, we consider its corresponding genetic pair $(x_g^i)$ as the positive sample with all other genetic samples $(x_g^j; j \neq i)$ as negatives to be contrasted against. Similarly, in (2)—genetics-to-imaging, imaging $h_v^i$ is the positive of the genetic sample $x_g^i$ and contrasted against all other images in the batch $(h_v^j; j \neq i)$. In (3)—imaging-to-imaging, and (4)—genetic-to-genetic loss terms, by taking advantage of the labels, we use intra-modal contrasting to align the representations of the samples of same class to be close to each other, while the samples from the other class are pushed apart. Finally, we combine the four loss terms and define the contrastive batch loss function.

$$\mathrm{L}(v, g) = -\sum_{i=1}^{I} \log \frac{\exp\left(\frac{z_v^i \cdot z_g^i}{\tau}\right)}{\sum_{j=1, j\neq i}^{I} \exp\left(\frac{z_v^i \cdot z_g^j}{\tau}\right)} \quad (1)$$

$$\mathrm{L}(g, v) = -\sum_{i=1}^{I} \log \frac{\exp\left(\frac{z_g^i \cdot z_v^i}{\tau}\right)}{\sum_{j=1, j\neq i}^{I} \exp\left(\frac{z_g^i \cdot z_v^j}{\tau}\right)} \quad (2)$$

$$L(v,v) = \sum_{i=1}^{I} \frac{-1}{|P(i)|} \sum_{p \in P(i)} \log \frac{\exp\left(\frac{z_v^i \cdot z_v^p}{\tau}\right)}{\sum_{j=1, j \neq i}^{I} \exp\left(\frac{z_v^i \cdot z_v^j}{\tau}\right)} \quad (3)$$

$$L(g,g) = \sum_{i=1}^{I} \frac{-1}{|P(i)|} \sum_{p \in P(i)} \log \frac{\exp\left(\frac{z_g^i \cdot z_g^p}{\tau}\right)}{\sum_{j=1, j \neq i}^{I} \exp\left(\frac{z_g^i \cdot z_g^j}{\tau}\right)} \quad (4)$$

$$L_{cont}(v,g) = \lambda L(v,g) + \sigma L(g,v) + \gamma L(v,v) + \theta L(g,g) \quad (5)$$

Where $\lambda, \sigma, \gamma, \theta \in [0,1]$ are weighting hyperparameters. By incorporating these inter- and intra-modal contrastive losses, our model ensures that each modality independently learns meaningful and discriminative representations. These representations are then combined for the final classification task, enhancing the overall performance of the model. In addition, we trained the model with different combinations of these loss terms and compare the quality of the representations by fine-tuning to the classification task shown in Stage 3.

*3) Classification of WM*

In stage 3, we combined genetic representations obtained from the contrastive training ( $z_g^i$ ) with the imaging representations from the 3D CNN model ($h_v^i$), resulting in an input dimension of 612 (100 from genetic encoder and 512 from the imaging CNN) and trained a fully-connected neural network. The neural network has three hidden layers of size 250, 200, and 150, respectively. The output layer has two neurons for classification of WM.

*E. Baseline models*

We compare the performance of our model with the following four baseline models. For the first two models, we trained two supervised models for each modality separately. Next, we combined the latent representations from the individual modalities and trained iii) a linear classifier using Support Vector Machine (SVM), and iv) a fully connected neural network classifier.

**Imaging CNN**: For classification of WM using only the imaging modality we used the same model described in the stage 1 of the CGNI framework.

**Sparse Genetic Classification Models**: The genetic data was processed using a fully connected neural network (NN) designed to capture the complex relationships within the 1060 SNPs. The architecture of the NN is as follows: the input layer of the network takes an input of 1060 SNP features. The hidden layers are of size 800, 600, and 400 neurons respectively. The final layer is a 2-neuron output layer to classify the WM scores. To prevent overfitting and ensure sparsity, L1 regularization was applied to the hidden layers. The activation function used for the hidden layers is the ReLU. We denote the output of the last hidden layer (size of 400 neurons) as $h_g^i = f_g(x_g^i) \in \mathbb{R}^{400}$ , where $x_g^i$ is the genetic modality of the participant i $\in \{1, ..., 4995\}$.

**Imaging-Genetic Integration Models (linear and non-linear)**: Finally, we combined the output embeddings of the two modalities, $h_v^i \oplus h_g^i$ , and trained a Support Vector

Machine (SVM) for linear classification and a fully connected NN. The input layer of this NN has 912 neurons. The input layer is followed by two hidden layers of size 500 and 200 neurons. The output layer has 2 nodes corresponding to high memory or low memory. ReLU activation was used after each layer to incorporate non-linearity into the model. While training, L1 regularization was applied to all the layers to prevent overfitting.

*F. Post Analyses for results interpretation*

In our post analyses, first, we utilized Sparse Canonical Correlation Analysis (SCCA) to identify the relationships between the imaging and genetic latent representations, and top contributing features. Next, from the identified genetic features, we further performed gene enrichment analysis to reveal their biological significance.

*1) Sparse CCA Analysis for Understanding Imaging-Genetics Relationships*

To understand the relationships between the imaging and genetic representations of the contrastive training, we performed SCCA on the encoder outputs $z_v^i, z_g^i$. We utilized the iterative penalized SCCA method proposed by Mai et. al. [28] due to its ability to handle high-dimensional and enforce sparsity to reduce overfitting. We considered the number of components as one of the hyperparameter along with the sparsity penalization for each modality and selected the sCCA model via GridSearchCV and 5-fold cross validation. We used the implementation provided by Chapman et. al. [29].

For the correlated components from the imaging and genetic modalities, to identify the top contributing important features, we used feature occlusion sensitivity method. We denote the i[th] sample with k[th] feature occluded as $x_{mk}^i$, where $m$ is the modality. The feature occluded imaging encoder $z_{vk}^i = e_v\left(f_v(x_{vk}^i)\right)$ and $z_{gk}^i = e_g(x_{gk}^i)$ as the genetic encoder output. For imaging modality, we partitioned the brain into 116 brain regions based on [30]. For genetic modality, we considered the individual SNPs as features and computed their contribution. Given for each correlated component identified from sCCA, only fewer latent representation nodes are involved. We computed the contribution of each feature to the largest weighted (highest absolute canonical weight) representation node. The contribution value is computed as the mean difference of the actual encoder output $z_m^i$), and encoder output with feature occluded$z_{mk}^i$) as shown in (6).

$$contribution_k = \frac{\sum_{i=0}^{N} z_m^i - z_{mk}^i}{N} \; \forall \, k \in features \quad (6)$$

The top contributing features to each component are the brain regions and SNPs whose contribution scores are greater than 2.5 and 3 standard deviations away, respectively, from the mean contribution scores.

*2) Gene Set Enrichment Analysis*

To further understand the biological significance of the set of SNPs that are identified to have significant contribution to the sCCA components, we have performed a gene enrichment analysis using the gProfiler online software [31]. First, we matched the SNPs to the Ensemble ID of the genes (these were part of the eQTL dataset obtained from

psychEncode) and used the g:Convert function to get the gene names, then used g:GOSt function to perform the enrichment test [31].

## G. Experimental Setup

Across all experiments, we used the same splits of data for training, validation, and test set. We ensured there was no cross-contamination of samples between the splits. We split the data into two sets, 90% for training and 10% for test set. Using the training set we performed five-fold cross validation for each set of hyperparameters. For fully-connected networks, we vary the number of hidden layers and the size of each layer, learning rate. We used the Adam optimizer for optimizing the neural networks. For the CNN models we used a batch size of 25 images on A100 GPU. For the contrastive model we used a batch size of 50 and for the WM classification model the batch size is 75. For classification tasks we used balanced accuracy as our metric.

## III. RESULTS

### A. Baseline Model Results

As a benchmark comparison for the contrastive training model, we have reported the performance of several baselines. In TABLE 1, we report the mean balanced accuracy across five-folds of the imaging-genetic baseline model along with other models trained on individual modalities.

TABLE 1
COMPARISON OF BALANCED ACCURACIES OF MODELS

| WM Classification | Training | Validation | Holdout |
|---|---|---|---|
| Imaging CNN | 81.34 ± 2.34 | 75.32 ± 0.49 | 87.02 ± 1.31 |
| Sparse Genetic NN | 59.80 ± 0.42 | 59.35 ± 0.45 | 59.28 ± 0.57 |
| Imaging-Genetic SVM (linear) | 95.5 ± 2.59 | 65.08 ± 1.56 | 65.23 ± 2.38 |
| Imaging-Genetic NN (non-linear) | 87.47 ± 0.45 | 87.08 ± 0.80 | 87.02 ± 0.67 |
| Imaging-Genetic contrastive learning (proposed with best terms) | 91.37 ± 0.30 | 90.39 ± 0.11 | 88.91 ± 0.66 |

To pre-train the CNN model for brain age prediction task, we achieved the best results (MAE of $2.82 \pm 0.04$ on training, $2.88 \pm 0.05$ on validation, $2.47 \pm 0.2$ on test set). By transferring the first two layers of the brain age model, we achieved high mean balanced accuracy (87.02) on the holdout set. For genetic modality, With the addition of L1 sparsity explicitly and L2 sparsity via the weight decay parameter of the Adam optimizer the model achieved balanced accuracies around 59% Finally, the multi-modal imaging-genetic model exhibited the highest performance with balanced accuracies above 87% across all splits, indicating robust generalization.

## B. Comparison of Combinations of Contrastive Loss Terms

We systematically tested various combinations of loss terms in (5) and recorded the balanced accuracies on the training, validation, and test sets. The results are summarized in the Table 2.

Table 2
BALANCED ACCURACIES ACHIEVED FOR DIFFERENT LOSS TERM COMBINATIONS

| Loss Term | Train | Validation | Holdout |
|---|---|---|---|
| Imaging-to-genetics (L1) | 89.96 ± 0.80 | 88.55 ± 1.10 | 88.31 ± 0.31 |
| Genetics-to-imaging (L2) | 89.81 ± 0.95 | 87.97 ± 0.91 | 87.51 ± 0.20 |
| Imaging-to-imaging (L3) | 88.99 ± 0.74 | 87.45 ± 1.60 | 86.80 ± 0.34 |
| Genetics-to-genetics (L4) | 90.03 ± 0.87 | 88.04 ± 0.76 | 88.1 ± 0.41 |
| L1 + L2 | 88.98 ± 1.13 | 88.03 ± 1.26 | 87.76 ± 0.94 |
| L1 + L3 | 90.09 ± 1.1 | 88.14 ± 0.98 | 88.19 ± 0.67 |
| L1 + L4 | 89.75 ± 1.21 | 89.33 ± 0.11 | 88.01 ± 0.08 |
| L2 + L3 | 90.45 ± 0.37 | 87.89 ± 0.87 | 89.66 ± 0.31 |
| L2 + L4 | 90.49 ±0.84 | 88.27 ± 1.31 | 88.07 ± 0.83 |
| L3 + L4 | 88.95 ± 1.71 | 87.96 ± 0.86 | 87.74 ± 0.43 |
| L1 + L2 + L3 | 89.60 ± 0.98 | 88.22 ± 1.03 | 88.18 ± 0.34 |
| L2 + L3 + L4 | 90.00 ± 1.09 | 88.14 ± 1.01 | 87.79 ± 0.33 |
| L1 + L2 + L4 | 91.37 ± 0.30 | 90.39 ± 0.11 | 88.91 ± 0.66 |
| L1+ L2+ L3+ L4 | 89.81 ± 0.73 | 88.38 ± 0.99 | 88.61 ± 0.56 |

All individual loss terms have shown improvement over the baseline accuracies. The combination of loss terms (1), (2), and (4) yielded the highest balanced accuracies on the training ($91.37 \pm 0.30\%$), validation ($90.39 \pm 0.11\%$), and test sets ($88.91 \pm 0.66\%$). The values of the loss term weighting hyperparameters ($\lambda, \sigma, \theta$) for each of the loss term were 0.7, 0.3, and 0.4, respectively. Individual loss terms (1) and (4) also performed well, particularly in the holdout sets ($88.31 \pm 0.31\%$ and $88.10 \pm 0.41\%$, respectively).

## C. Post Analysis Results

The SCCA analysis performed on the representations of the contrastive model with loss terms (1), (2), and (4) using the GridSearchCV resulted in selection of five components. In

Table 3, we reported the correlation coefficients for these five component pairs across the training, validation, and holdout sets. The correlations for each component pair are relatively consistent between the training and validation sets, indicating the stability of the CCA model. The top contributing brain regions and SNPs are listed in Table 4 and

Table 5, and plotted in Fig. 3. Brain regions in red indicate increased effect and blue regions indicate decreased effect.

#### Table 3
CCA CORRELATION OF COMPONENTS

| Component | Training | Validation | Holdout |
|---|---|---|---|
| 1 | 0.68 | 0.67 | 0.57 |
| 2 | 0.67 | 0.65 | 0.54 |
| 3 | 0.66 | 0.65 | 0.55 |
| 4 | 0.65 | 0.65 | 0.52 |
| 5 | 0.63 | 0.63 | 0.57 |

Across five SCCA components we identified seven brain regions and the 28 genes corresponding to the identified SNPs. Most of the selected regions and SNPs are similar among all the components.

#### Table 4
GENES WITH SIGNIFICANT CONTRIBUTION SCORES FOR EACH COMPONENT

| Component | Genes |
|---|---|
| 1 | ADAM10, LRRK2, RAB6A, GSK3B, BTBD1, RBM11, BIN1, FIS1, RNH1, STAU1, CD9 |
| 2 | ADAM10, LRRK2, RAB6A, ATP9A, BTBD1, MAPT, CALHM1, RPL28, FIS1, STAU1, GSE1, CD9 |
| 3 | ADAM10, LRRK2, RAB6A, BTBD1, MAPT, RPL28, RCAN1, FYN, KLHL35, STAU1, CD9 |
| 4 | MFF, RAB6A, BTBD1, BIN1, CALHM1, FIS1, SLC40A1, CENPV, RNH1, STAU1 |
| 5 | LRRK2, ATP9A, BTBD1, CALHM1, FIS1, CCNA1, SLC40A1, RNH1, STAU1 |

#### Table 5
BRAIN REGIONS WITH SIGNIFICANT CONTRIBUTION SCORES FOR EACH COMPONENT

| Component | Brain Regions |
|---|---|
| 1 | Left Cerebellum, Left Insula, Left Putamen, Left Caudate |
| 2 | Right Occipital Inferior, Left Caudate, Right Cuneus |
| 3 | Left Cerebellum, Left Insula, Left Caudate |
| 4 | Left Temporal Middle, Right Cuneus, Left Caudate |
| 5 | Right Occipital Inferior, Right Cuneus, Left Caudate |

#### D. Gene Enrichment Pathway Analysis

The gene enrichment test in three GO categories revealed enrichment in dendrite (p = 9.69e-6), synapse (p= 5.08e-5), and exocytic vesicle (p = 3.87e-4) in cellular components; regulation of mitochondrial fission, and developmental process in biological pathways, and protein binding in molecular function. The results of the analysis can be found here (https://biit.cs.ut.ee/gplink/l/H8Pp77ilQy).

## IV. DISCUSSION

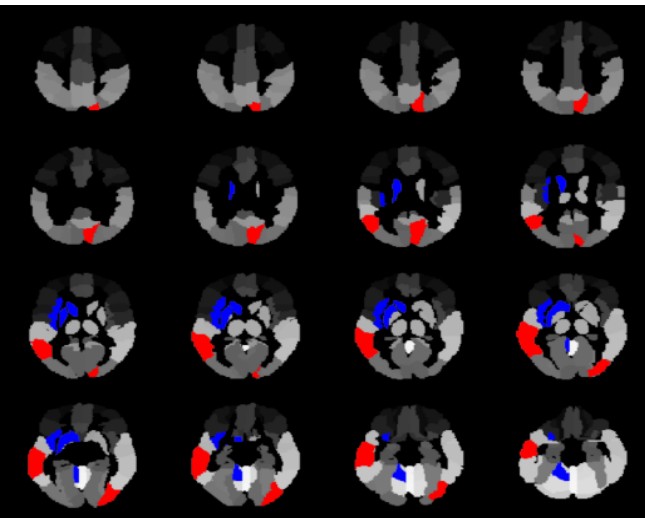

Fig 3: Significant brain regions identified across five SCCA components

This study aimed to integrate genetic and imaging data using a novel contrastive learning framework to identify significant associations between genetic variants and brain regions associated with cognitive function of WM. Our comprehensive analysis, which included transfer learning contrastive learning, SCCA, and gene enrichment tests, yielded several important findings.

The performance of various models on WM classification tasks demonstrated that integrating imaging and genetic data outperformed models using only one data type. The Imaging-genetic neural network achieved the highest balanced accuracy (87.02 ± 0.67) on the holdout set, indicating the potential of multi-modal approaches in classification of WM. The study of different combinations of the loss terms further showed that including terms (L1, L2, L4) resulted in the best performance, highlighting the importance of capturing relationships between both modalities. The inclusion of (4) in the loss function (i.e., the loss term L(g,g)) showed substantial improvement in validation accuracy (89.33 ± 0.11%) compared to other combinations, indicating its importance for generalization. This combination effectively leverages the complementary information from both imaging and genetic data, resulting in superior performance in WM classification tasks.

Our post analysis performed to identify the associations between brain regions and genes indicate that a decrease in the Grey Matter Volume (GMV) of regions left cerebellum, left insula, left putamen, left Caudate, and increase in the GMV of regions right occipital inferior, right cuneus, and left temporal middle is associated with the increase in the minor allele in the SNPs of all identified genes except gene STAU1. Recent research has shown that the reductions in cerebellar volume accompanying aging and are correlated with cognitive decline [32] which underscores our finding regarding left cerebellum in relation to working memory in older population.

The gene enrichment analysis highlighted several biological processes and cellular components significantly associated with the identified genes. The identified SNPs and associated genes provide insights into the genetic

underpinnings of cognitive function. Notably, the regulation of mitochondrial fission and developmental processes were prominent, suggesting that these pathways may play crucial roles in maintaining WM capacity and preventing neurodegeneration. Recent research has identified several genetic variants that significantly impact WM performance. For instance, genes like ADAM10, LRRK2, RAB6A, and BTBD1 have been implicated in various neural processes relevant to WM. ADAM10, involved in amyloid precursor protein processing, has been linked to Alzheimer's disease and cognitive decline [7]. Variants in LRRK2, associated with synaptic vesicle trafficking, also play a role in neurodegenerative diseases such as Parkinson's, influencing cognitive functions [33]. The integration of these genetic data with neuroimaging findings helps elucidate the complex interplay between genetic predispositions and brain structure in maintaining WM. While our study provides valuable insights, several limitations should be noted. The sample size, while sufficient for the current analysis, limits the generalizability of the findings. Larger, more diverse cohorts are needed to validate these results. The identified genetic variants and pathways require further functional validation to establish causal relationships with WM capacity.

## V. Conclusion and Future Work

In conclusion, our study demonstrates the power of integrating genetic and imaging data using contrastive learning techniques. The identified genetic variants and brain regions, along with their associated biological pathways, provide a foundation for further exploration into the genetic basis of WM. This integrated approach holds promise for advancing our understanding of the complex interactions underlying WM capacity and its decline in neurodegenerative diseases.

To further validate our findings, we plan to apply the model to other large-scale biobanks such as the Human Connectome Project which encompass diverse populations and different environments, will allow us to assess the generalizability of our model. We also envision that with continued development, our model could predict not only WM capacity but also the progression of memory performance over time. This information could be valuable in clinical settings, where it could guide treatment plans by identifying individuals at higher risk of memory decline. Moreover, as contrastive learning models improve, they can help handle the challenges of applying the model to new cohorts from different hospitals, where sequencing or imaging methods may vary.

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
