# OpenReview forum: "Integrating Neuroimaging and Genetics via Contrastive Learning for Working Memory"
_IEEE.org/EMBS/BHI/2024/Conference — IEEE BHI'24_

### Official Review · Reviewer_FgfZ · 2024-08-11
**Data integration constrastive learning framework for improving performance of working memory classification schemes**

**Overall Rating:** 7
**Confidence:** 5

**Other Quality Metrics:**

(a) Clarity of writing                : GREAT, paper is well-organized and written in a fluent language
(b) Clinical Significance          : GOOD, potential of proposed system for application in clinical scenarios is high, yet additional data are required considering datasets of increased size and incorporating medical expert ground truth information in the model training procedure
(c) Methodological Novelty     : GOOD, innovative characteristics of presented study are mostly related to the application field and the integration of data sources adopting an existing contrastive learning model. Technical details on innovative characteristics introduced in discrete algorithmic stages could be further indicated
(d) Experiments and Results  : GOOD, since quantitative and comparative assessments are performed, revealing improved performance of learning models upon integrating neuroimaging and genetics data. Additional validation on wider datasets and engaging medical experts is required to ensure robustness and precision of developed methodology

**Questions For The Authors:**

In order to further improve the quality of the manuscript, the following amendments could be addressed:
* what is the motivation beyond the selection of specific baseline models for extracting comparative results?
* is post analysis validated by medical experts?
* how could proposed framework be adopted to real clinical practice scenarios?

**Strengths:**

Advantages of presented study can be summarized as follows:
* clear indication of paper novelty and contribution
* precise description of implementation scheme, which is supported by informative figures and proper theoretical background to enable reproducibility of results by any individual researcher
* detailed review of related work based on numerous and recent references, revealing limitations and challenges with respect to the examined scientific field
* extended validation on real benchmark dataset, quantitative and comparative assessments are included
* potential to identifying factors related to brain structure/activity and correlating to other neuro-diseases

**Summary Of The Paper:**

Presented study introduces a contrastive learning framework for fusing neuroimaging and genetics data towards improving the classification of high vs. low working memory capacity and identifying factors associated with brain structure. Based on a benchmark dataset, the authors validated proposed approach and proved that integrating different kinds of data improves performance of working memory classification schemes.

**Weaknesses:**

Weaknesses of presented study can be summarized as follows:
* additional data are required in order to generalize outcome of proposed approach, identified genetic variants and pathways require further clinical input and ground truth information in order to reveal significant correlation to working memory capacity

---

### Official Review · Reviewer_ibVZ · 2024-08-11
**Integrating Neuroimaging and Genetics via Contrastive Learning for Working Memory**

**Overall Rating:** 6
**Confidence:** 3

**Other Quality Metrics:**

Clarity of writing: good.
Clinical Significance: great
Methodological Novelty: good
Experiments and Results: fair

**Questions For The Authors:**

Would it be possible to rewrite thinking for readers in mind who are not familiar with the genetics and/or the functional MRI knowledge?

**Strengths:**

The proposed design of the model outperforms baseline models.

**Summary Of The Paper:**

This paper discusses a new design for multi-model learning for classification of worker memory functions using genetic SNP data and functional brain MRI images.

**Weaknesses:**

The proposed design of the model is a combination of multiple methods and it is not clear how much of it belongs to previous work and how much is novel.

It is difficult to read the experimental study part. It must be made clear which result is the result of the proposed model.

---

### Official Review · Reviewer_EpHc · 2024-08-14
**A good approach to study to improve the genetic-imaging relationships**

**Overall Rating:** 7
**Confidence:** 5

**Other Quality Metrics:**

Clarity of Writing: Good
Clinical Significance: Fair
Methodological Novelty: Excellent
Experiments and Results: Great

**Questions For The Authors:**

1.	Can you write a new section for data preprocessing?  I believe readers would benefit from a separate section than II-A, with more of the computational aspects of the way the data is processed, rather than the descriptions in II-A. Also, please touch upon the data normalization as well.
2.	What kind of external validation can be done to further highlight the predictive performance as well as the gene signatures revealed? Are there any other studies that have attempted something similar?
3.	While giving good performance due to the loss associated with contrasting genetic and image features, contrastive models are generally harder to train and to generalize to new populations without matching data points. Here, for a new cohort from a new hospital, what are the steps that you can identify to ensure a generalizable system? I’ve seen many models that work on one cohort but fails miserably in data from a new sequencing or imaging device or from another hospital that handles their data differently. In essence, what kind of real-world clinical relevance does your model have?

**Strengths:**

1. Contrastive learning-based framework to integrate genetic and neuroimaging data is interesting. It shows promise, which is supported by the results later.

2. Identifying associations between genetic variants and brain structure is crucial in further enhancing our understanding of AD and similar diseases. So a methodology under this umbrella is always welcome.

**Summary Of The Paper:**

The paper titled "Integrating Neuroimaging and Genetics via Contrastive Learning for Working Memory" aims to advance the understanding of the genetic and neural bases of working memory (WM) by integrating single nucleotide polymorphism (SNP) and neuroimaging data from the UK Biobank. The authors focus on improving the classification of high vs. low working memory capacity and identifying genetic factors associated with brain structure. The study utilizes 1060 SNPs related to amyloid precursor protein and Alzheimer's disease and integrates them with latent features of whole-brain gray matter density extracted using a pre-trained convolutional neural network (CNN). The integration is achieved through a novel supervised contrastive learning model, which enhances genetic-imaging relations within individuals and working memory groups. The study finds that features derived from contrastive learning outperform baseline models in classification tasks.

**Weaknesses:**

~ This is supposed to be a blind submission. What about the link provided in the last section: "The results of the analysis can be found here (https://biit.cs.ut.ee/gplink/l/H8Pp77ilQy)."

---

### Decision · Program_Chairs · 2024-09-23

Accept